# Transcriptomic Analysis across Crayfish (*Cherax quadricarinatus*) Claw Regeneration Reveals Potential Stem Cell Sources for Cultivated Crustacean Meat

**DOI:** 10.3390/ijms25168623

**Published:** 2024-08-07

**Authors:** Lisa Musgrove, Avani Bhojwani, Cameron Hyde, Susan Glendinning, Josephine Nocillado, Fraser D. Russell, Tomer Ventura

**Affiliations:** 1Centre for Bioinnovation, University of the Sunshine Coast (UniSC), 4 Locked Bag, Maroochydore, QLD 4558, Australia; lisa.musgrove@research.usc.edu.au (L.M.);; 2School of Science, Technology and Engineering, University of the Sunshine Coast (UniSC), 4 Locked Bag, Maroochydore, QLD 4558, Australia; 3Queensland Cyber Infrastructure Foundation (QCIF) Ltd., The University of Queensland, St. Lucia, QLD 4072, Australia; 4School of Health, University of the Sunshine Coast (UniSC), 4 Locked Bag, Maroochydore, QLD 4558, Australia

**Keywords:** cultivated crustacean meat, limb regeneration, stem cells, RNA-Seq, differential gene expression, cell proliferation, muscle development

## Abstract

In the face of rising global demand and unsustainable production methods, cultivated crustacean meat (CCM) is proposed as an alternative means to produce delicious lobster, shrimp, and crab products. Cultivated meat requires starting stem cells that may vary in terms of potency and the propensity to proliferate or differentiate into myogenic (muscle-related) tissues. Recognizing that regenerating limbs are a non-lethal source of tissue and may harbor relevant stem cells, we selected those of the crayfish *Cherax quadricarinatus* as our model. To investigate stem cell activity, we conducted RNA-Seq analysis across six stages of claw regeneration (four pre-molt and two post-molt stages), along with histology and real-time quantitative PCR (qPCR). Our results showed that while genes related to energy production, muscle hypertrophy, and exoskeletal cuticle synthesis dominated the post-molt stages, growth factor receptors (FGFR, EGFR, TGFR, and BMPR) and those related to stem cell proliferation and potency (Cyclins, CDKs, Wnts, C-Myc, Klf4, Sox2, PCNA, and p53) were upregulated before the molt. Pre-molt upregulation in several genes occurred in two growth peaks; Stages 2 and 4. We therefore propose that pre-molt limb regeneration tissues, particularly those in the larger Stage 4, present a prolific and non-lethal source of stem cells for CCM development.

## 1. Introduction

### 1.1. Cultivated Crustacean Meat

The cultivation of crustacean meat directly from stem cells may allow us to produce delicious seafood without having to farm or fish animals such as lobster, shrimp, crabs, and crayfish. As global demand for these products grows [1], aspects of their current production have become more concerning. Many wild-caught crustacean enterprises are responsible for over-fishing and by-catch in the seas, and products can be contaminated with heavy metals or microplastics [2,3,4,5]. Although intended as a more sustainable alternative, aquaculture farming of crustaceans is also faced with environmental challenges. It has been linked to excessive greenhouse gas production and coastal effluent pollution and is increasingly vulnerable to disease outbreaks and the warming climate [1,6,7,8,9,10]. Furthermore, as crustaceans are becoming recognized as sentient, current fishing and farming methods are unlikely to meet emerging animal welfare standards [11,12,13]. Alternative and complementary production methods, such as cell cultivation, are needed to ensure we have a more sustainable and ethical global supply of these foods.

### 1.2. Stem Cells and Myogenic Factors

The development of cultivated meat (CM) requires careful selection of starting stem cells which need to proliferate sufficiently and differentiate into meat-relevant tissues such as muscle and fat. Stem cells that are commonly targeted for CM production range in their ability to proliferate and differentiate and include pluripotent stem cells (PSCs) such as embryonic or induced PSCs, multipotent mesenchymal stem cells, and fibro-adipogenic progenitors, or unipotent muscle stem cells (MSCs) including myoblasts and satellite cells [14,15,16,17]. Like vertebrates, crustaceans possess a diverse array of stem cells that are likely to range in their proliferative and myogenic abilities [18]. To date, no crustacean cell lines have been developed, so cultivated crustacean meat (CCM) research must rely on primary sources of stem cells, which, due to the crustacean exoskeleton, can be difficult to obtain non-lethally [15]. As the development of CM seeks to reduce reliance on animals and improve animal welfare outcomes, avoiding euthanasia as practicably as possible is warranted. Therefore, we have decided to investigate the regenerating limb as a non-lethal source of stem cells for CCM.

### 1.3. Regenerating Limb Bud Model

Utilizing a process called autotomy, many crustaceans can eject their limbs at a preformed breakage plane (Figure 1) with minimal blood loss or tissue damage and regrow them to a pristine state over successive molt events. Underneath the breakage plane, a blastema forms from migrating and dedifferentiating cells and subsequently undergoes epimorphic regeneration, which is thought to be replicative of embryogenesis [19,20,21,22,23]. A recent analysis, however, suggests that there are distinct temporal differences between primordial and regenerative limb development [24]. Nonetheless, crustacean limb regeneration clearly involves the proliferation of undifferentiated (PSC-like) stem cells and their subsequent myogenic differentiation into MSCs and muscle tissue. Identifying when pluripotent (or multipotent) and myogenic stem cell activity is more prevalent during the limb regeneration process would be useful for CCM research and starting cell selection.

### 1.4. Molecular Factors Relevant to CM and CCM Research

Numerous molecular factors related to stem cell proliferation and myogenic differentiation, including classic markers of PSCs and MSCs, have been identified as potentially relevant to CM and CCM research [15,16,17]. Compared to vertebrate models, there is considerably less known about these in crustacean species. To shed light on this, and to begin investigating the usefulness of the regenerating limb model for CCM development, we have conducted RNA-Seq analysis across six stages of claw regeneration in the Australian Redclaw crayfish, *Cherax quadricarinatus.* In addition to its commercial importance, robust nature, and ease of handling, this species was selected due to its well-established research background including a rapidly growing body of genomic, physiological, and behavioral data.

Although recent studies have analyzed gene expression across the crustacean limb regeneration process [24,25], to date, none have focused on genes specifically relevant to potential tissue sources for CCM development. To elucidate this, we present here first a broad-spectrum analysis of differential gene expression across the six stages to determine if any prominent molecular changes relate specifically to stem cell activity, such as cell proliferation or classic markers of PSCs and MSCs. We also present a relative expression analysis of key target genes identified as endocrine factors, myogenic factors, cell cycle factors, and pluripotency factors. After isolating two particular regeneration stages as worthy of further investigation, we conducted histology and real-time quantitative PCR (qPCR) to compare evidence of stem cell activity between the two stages. Our findings demonstrate clear differences between all stages, highlighting those with molecular activity indicative of cell types suited to CCM development [15].

## 2. Results

### 2.1. Stage Allocation and Sample Preparation

Based on limb regeneration studies of the fiddler crab *Uca pugilator* [19,20,26] and the crayfish *Cherax destructor* [27], as well as our own observations of morphology and allometric growth, regenerating and regenerated claws of *C. quadricarinatus* adults were grouped into one of six stages: Stage 1—post autotomy with no observable growth; Stage 2—the emergence of an unsegmented papilla; Stage 3—beginnings of segmentation with a closed dactyl; Stage 4—almost fully segmented half-size limb immediately pre-molt; Stage 5—a full-sized post-molt limb with fully segmented but soft cuticle; and Stage 6—the cuticle-hardened and fully regenerated intermolt limb (Figure 1). RNA was extracted from each regenerated or full claw, and the quality was assessed. Two of the 24 RNA samples failed initial library preparation quality control. Individual samples, their stages, R-values, and QC status are detailed in Table 1.

### 2.2. Transcriptome Sequencing, Assembly, and Quantification

A total of 22 RNA-Seq libraries were sequenced with Illumina NextGen sequencing, resulting in a minimum of 39 million cleaned reads per library. Six of these libraries (one per stage) were trimmed, then assembled de novo and clustered into a transcriptome resulting in 55,018 transcripts. Quantification was performed at the transcript level and then aggregated to 29,542 gene-level counts, which were subsequently normalized. BUSCO scores are presented in Table 2. The assembled transcriptome, quantification output files, and normalized gene counts are provided in Appendix A, respectively.

### 2.3. Annotation of the Reference Transcriptome

Annotation of the initial reference assembly was conducted using top BLASTx hits from SwissProt (https://www.uniprot.org/) (Accessed 1 May 2023). This resulted in 39, 360 ORFs with Gene Ontology (GO), Evolutionary Genealogy of Genes: Non-supervised Orthologous Groups, Clusters of Orthologous Genes (eggnog COG), Kyoto Encyclopedia of Genes and Genomes (KEGG) pathways, and protein family (PFAM) domains and/or BLAST annotations (Table 3). The annotation output file is provided in Appendix A.

### 2.4. Principal Component Analysis

A principal component analysis was conducted on the top 500 genes, selected by highest row variance (Figure 2). While showing clear grouping for Stages 1, 5, and 6, Stages 2 and 3 shared some cross-over, and Stage 4 included a single outlier.

### 2.5. Selection of Differentially Expressed Genes

Transcript-level quantification was aggregated to gene-level abundance before differential expression analysis was conducted on all 15 pair-wise comparisons among the six regeneration stage groups, resulting in 16,450 differentially expressed genes (DEGs). Of those with SwissProt annotations, the top ten upregulated and top ten downregulated DEGs in each comparison group were collated and cleared of duplicates. Normalized gene counts for DEGs with the same gene name were averaged, as were sample counts for each regeneration stage. This resulted in 92 unique annotated DEG records which were plotted in a heatmap (Figure 3). A list of all DEG transcripts, their gene counts, statistical scores, and SwissProt annotations for each of the 15 pair-wise comparisons is found in Appendix A.

### 2.6. GO Enrichment of Differentially Expressed Genes

The GO term annotations of the DEGs were subject to enrichment analysis, with the 10 most significant annotations for each comparison (node size 70) returning 227 biological process (BP), 126 cellular compartment and 123 molecular function GO terms. Significantly enriched annotations of DEGs with all three GO term categories can be found in Appendix A. The 227 BP GO terms were curated to exclude duplicates and uninformative terms. The relative expression of the resulting 53 enriched BP GO terms, across six regeneration stages, was calculated by summing the normalized gene counts of all contigs contributing to that GO term (Figure 4).

### 2.7. Functional Characterization of Differentially Expressed Genes and GO Terms

Cross-checking of the gene functions at www.uniprot.org (Accessed 27 May, 2024) and www.ncbi.nlm.nih.gov (Accessed 27 May, 2024) enabled the grouping of the top 92 annotated DEGs into 12 functional categories, and the 53 enriched GO terms into 9 functional categories (Table 4). GO term categories were based on functions inferred from the GO terms themselves, as well as annotation of the DEGs contributing to the GO terms. Merging the two category lists resulted in 14 functional categories. Normalized gene counts of all transcripts in each category were averaged, and the relative expression of each category was plotted across the six stages (Figure 5). The six categories evident in both analyses were ‘Neuronal development’, ‘Wound repair/immune response’, ‘Tissue regeneration’, ‘Energy production’, ‘Metabolism/metabolite production’, and ‘Muscle development’. Those unique to the top 92 DEGs are ‘Cuticle development’, ‘Pigmentation’ (blue and red), and ‘Cell repair/homeostasis’, and those unique to the GO enrichment analysis were ‘Cell cycle activity’, ‘Insulin-related growth’, and ‘Protein translation’.

The functions that dominated in the earliest stages were wound repair/inflammation, tissue development, and cell repair. Those mostly in the post-molt stages were muscle development, protein translation, energy production, metabolite production, CP-type cuticle proteins (related to hard exoskeletons) [28], and red pigmentation. In the immediate pre-molt stage (Stage 4), AM/AMP-type cuticle proteins (related to flexible, membranous cuticles) [28] and blue pigmentation dominated.

Wound repair and immune response genes showed higher expression in the earliest stages. Tissue development and cell repair/homeostasis genes showed similar expression. Neuronal development and cell cycle activity showed continual expression across the regeneration process albeit with less in the post-molt stages, particularly for the cell cycle genes. Upregulation in the middle stages (mostly Stage 4) was dominated by arthrodial membrane cuticle proteins (AM/AMP type) and blue cuticle pigmentation. In contrast, structural cuticle proteins (CP type) were expressed in the post-molt stages when the exoskeleton was hardening. The majority of muscle fiber development is upregulated after the molt when the newly enlarged limb fills with muscle tissue. In line with this intense anabolic activity, both metabolite synthesis and energy production are upregulated at the same time. Interestingly, there seemed to be a high presence of genes related to insulin metabolism and growth which occurred mostly at both ends of the regeneration process.

### 2.8. Identification and Relative Expression Analysis of Target Homologs

A list of target genes potentially relevant to CCM production from regenerating tissues was compiled based on our previous analyses [15] and grouped into four categories: endocrine factors, myogenic factors, cell cycle factors, and pluripotency factors. TransPi annotation results and BLAST searches with well-characterized orthologs of model organisms (human, mouse, and fruit fly) were used to identify target factors in the present transcriptome. Putative orthologs were then cross-checked for appropriate domains using SMART (http://smart.embl-heidelberg.de/) (Accessed 21 June, 2024) and relevant top BLAST hits on NCBI (https://blast.ncbi.nlm.nih.gov/Blast.cgi) (Accessed 21 June, 2024) and UniProt (https://www.uniprot.org/blast) (Accessed 21 June, 2024). A number of target factors could not be confidently identified (Table 5). Orthologs that were confidently identified were subject to further analysis. For the endocrine factors, relative expression across the six stages of limb regeneration was plotted in stacked curve graphs (Figure 6). One of these factors, the ecdysone receptor, was also subject to phylogenetic analysis (Figure 7). Relative expression was also plotted in stacked curve graphs for the myogenic factors (Figure 8), the cell cycle factors (Figure 9), and the pluripotency factors (Figure 10).

#### 2.8.1. Endocrine Factors

A number of endocrine factors and their receptors were identified with varying expression across the limb regeneration process (Figure 6). Most notably, the ecdysone receptor (EcR) and all growth factor receptors (except the insulin receptor) displayed a two-peaked expression pattern prior to the molt in Stages 2 and 4. Ligands BMP2 and TGFb shared reasonably continual expression with a gradual decline from a Stage 2 peak down to low expression in Stage 6. BMP1 by contrast, was more upregulated in Stage 5. EGF expression was greater in Stage 2 with some continuation in Stage 4 and none in the post-molt stages. The FGF homolog was upregulated in Stages 3 and 6 which is opposite to its supposed receptor. Numerous IGF-related proteins were identified with varying expression but overall concurring with the DEG analysis which showed insulin-related growth occurring in the earliest and latest stages. The only insulin-related receptor (IR) to be confidently identified showed greater upregulation in Stages 2 and 3. The PVF homolog showed similar expression to insulin-related genes (IGF/ILP and SIBD2). The PVF receptor (PVR), HGF, its receptor, and myostatin were not confidently identified in the current dataset.

Nuclear receptor families have recently been characterized in the tropical spiny lobster *Panulirus ornatus* [29] so there was an opportunity to further annotate the *C. quadricarinatus* EcR with phylogeny. Neighbor-Joining tree construction of receptor DNA binding domains showed that the *C. quadricarinatus* receptor clustered with the EcR family, providing further evidence of its expected function (Figure 7).

#### 2.8.2. Myogenic Factors

The myogenic factors investigated were those known to be involved in embryonic or adult myogenesis and myofibrillar proteins associated with hypertrophy. The *D. melanogaster* satellite cell marker, Zfh1, showed the most upregulation during the Stage 2 growth peak and some again during Stages 4 and 5. Clear homologs of the vertebrate satellite cell markers Pax 3 and Pax 7 could not be identified, but a Pax3/7-related protein PaxB1 showed upregulation across all stages except Stage 6, like Zfh1. The gene most commonly associated with myogenesis downstream from the Pax3/7 is MyoD and its *D. melanogaster* equivalent Nautilus (Nau). The closest Nau homolog in the current data showed clear upregulation in Stage 6. Mef2 which is known to work synergistically with other myogenic factors was primarily upregulated prior to the molt, peaking in Stage 4. In concurrence with the DEG analysis, all myofibrillar proteins were almost exclusively upregulated in Stage 6 with some slight uplift in Stage 4.

#### 2.8.3. Cell Cycle Genes

The cell cycle genes showed mostly coincidental expression which largely follows the two-peak pre-molt pattern observed for the endocrine receptors. Only CDK7, Cyclin A, Cyclin C, and the p53-inducible protein (P5I11) showed either upregulation or very minimal downregulation in the post-molt stages. The overall pattern concurs with the cycle activity observed in the DEG analysis (Figure 5).

#### 2.8.4. Pluripotency Factors

The pluripotency factors showed similar expression to the cell cycle genes which aligns with greater proliferative qualities of the more pluripotent stem cells. Most showed the same two-peak pattern to varying degrees, and all but Hedgehog were not expressed after the molt.

### 2.9. Stage 4 vs. Stage 6 Tissues as Potential Stem Cell Sources for CCM Development

The primary intention of this transcriptomic analysis was to identify optimal tissue sources of starting stem cells for CCM development. Full-grown intermolt claw tissue (akin to Stage 6) has been identified previously as a potential source of MSCs including satellite cells [31]; however, the present analysis suggests that tissues collected during pre-molt regeneration are undergoing more proliferative activity and may be a more viable source. Although such activity appears to peak equally in Stages 2 and 4, Stage 4 limbs are considerably larger and more accessible, so they are potentially a more productive source of stem cells. For this reason, Stage 4 tissues were selected for further comparative analysis against those of Stage 6.

#### 2.9.1. Histological Analysis

Hematoxylin and eosin and DAPI staining revealed some distinct differences between Stage 4 and 6 tissues (Figure 11). In the Stage 4 images, the new regenerating cuticle can be seen convoluted and compressed within the temporary cuticle, encasing what appears to be an outer layer of epithelial cells and immature muscle fibers within with many visible nuclei. Without confirmation through immunohistochemistry (IHC), it is hard to distinguish between myonuclei and putative MSCs in Stage 4 tissues. The Stage 6 tissue also showed an epithelial layer, although with much larger and less nucleated muscle fibers. There seemed to be a much higher ratio of nuclei to fibers in the Stage 4 tissues.

#### 2.9.2. Real-Time Quantitative PCR

To further validate the observed transcriptomic differences between Stage 4 and Stage 6 tissues, real-time quantitative PCR (qPCR) was conducted with five *C. quadricarinatus* genes: MLC (mature muscle protein), Nau (myogenesis marker), Oct3/4 (pluripotency marker), Zfh1 (putative satellite cell marker), and PCNA (cell cycle marker). The housekeeping gene Cq-18S was used as a positive control and to normalize the expression of the other genes. The qPCR trends largely reflected the RNA-Seq data with all but Zfh1 showing the same expression (Figure 12). Statistical tests revealed significant differences only in the Oct3/4 and PNCA samples; thus, more replicates will be needed to confirm the significance of the trends for the other target genes.

## 3. Discussion

Because crustacean limbs can regenerate, they offer a non-lethal and more accessible alternative to abdominal muscle or other internal crustacean tissues for CCM development [15]. Through our investigation of regenerating crayfish limbs, we found that pre-molt tissues harbor more stem cell activity than post-molt limb tissues, and of the pre-molt tissues, those from the later stages (Stage 4 in our study) may be the most practical and prolific source.

### 3.1. Differentially Expressed Gene (DEG) Analysis

A DEG analysis was conducted between all pair-wise comparisons among the six stages; that is, each stage was compared to every other stage to see if there were any standout patterns in gene expression. The results reveal distinct patterns of molecular and cellular activity across the six stages. Based on previous limb regeneration studies in *U. pugilator* [19,20,26] and *C. destructor* [27], we expected most wound healing and signs of PSCs to occur in the earlier stages, followed by initial myogenic determination and differentiation activity of MSCs, and then some mature muscle protein synthesis (hypertrophy) just prior to the molt event. Although increased cell cycle activity is a feature of PSCs [14,17], we expected mitosis to continue in party-differentiated stem cells throughout pre-molt tissue development. We expected that after the molt, all stem cell activity, including mitosis, would have ceased, and fully differentiated tissues would continue to grow via extensive myofibrillar protein synthesis (hypertrophy), which is required to fill the newly enlarged cuticle with muscle tissue.

This hypothesis was mostly borne out by the DEG results. Wound healing activity is upregulated in the earliest stages, and muscle hypertrophy predominates in the post-molt stages, with some uplift in the immediate pre-molt Stage 4. While cell proliferation appears to be uniformly present right up until the molt event. There is some uplift in Stage 6, though this is comparatively minimal. Neuronal development is seemingly required throughout the whole regeneration process, except for Stage 4. As discussed further below, Stage 4 is a high point for membranous cuticle development which may require less neuronal involvement compared to developing muscle tissue. However, there is still muscle development occurring in Stage 4, so this intriguing absence warrants further investigation.

Genes associated with tissue regeneration and cell repair follow the wound repair/inflammation pattern. This is supported by a well-established link between muscle tissue regeneration and the immune system in many animals [15]. In particular, a distinct role for immune-related hemocytes in crustacean muscle regeneration has previously been outlined, with a number of reports suggesting these cells may act as initiating stem cells in new muscle growth [21,23,32,33]. The coincidental upregulation of these three functional categories (wound repair, cell repair, and tissue regeneration) may therefore be directly related to hemocyte participation, suggesting that this cell type and the genes that drive their nuanced involvement should also be a point of focus for in vitro crustacean muscle development.

The functional categories of metabolite production, energy production, and protein synthesis all follow muscle development closely and can be easily explained by the increased energy and building blocks required for extensive structural protein synthesis occurring during the hypertrophic phase.

The insulin-related growth category is upregulated in the earliest and latest stages. This is likely due to the variety of roles for IGFs in tissue growth. For instance, different studies have shown that while both IGF1 and 2 have roles in myogenic differentiation and fusion, IGF1 is also required for the proliferation of myogenic stem cells [34,35,36]. A previous *C. quadricarinatus* study showed that IGF is correlated to increased overall body size and is required for the hypertrophic activity of muscle tissue explants [37] which could support the greater expression of insulin-related growth during Stage 6 hypertrophy. Furthermore, IGF1 and 2 were shown to significantly increase the proliferation of hemocyte precursors (hematopoietic stem cells) in the prawn *Penaeus monodon* [38]. If hemocyte activity is responsible for the Stage 1 upregulation of the wound repair, tissue regeneration, and cell repair categories mentioned earlier, this could also explain the concurrent uptick in insulin-related activity there.

We had not intended to highlight cuticle development or coloration as these are not directly related to growing muscle tissue in vitro; however, their expression was so dominating during certain stages of the limb regeneration process, so we opted to include them. During Stage 4, there were no other genes as differentially expressed than those relating to blue pigmentation and the soft, membranous AM/AMP cuticle protein types. As shown in Figure 1, this stage is when the regenerating limb begins to turn deep blue and is the largest of those with pliable, temporary cuticles, clearly explaining the upregulation of these gene categories. However, upon closer examination of the differential expression data, the upregulation of these in Stage 4 appears to be attributed to the single outlier sample; S4.2 (Figure 2). This limb regenerate had a much larger R-value (Table 1) and was thus larger and closer to the molt event and a deeper blue than the other three Stage 4 samples. In contrast, exoskeletal CP cuticle protein types are upregulated in the post-molt stages when the cuticle has hardened. The red pigmentation uptick in Stage 6 is interesting considering female *C. quadricarinatus* do not have any red coloration like their male counterparts. It is therefore likely the red pigment gene has alternative functions, either relating to cuticle development or other processes. Its significant differential expression in Stage 6 potentially warrants further investigation.

The overall pattern observed in the DEG analysis largely concurs with a recent analysis of limb regeneration in the amphipod *Parhyale hawaiensis* which determined that, unlike in embryonic limb development, the earliest stages are dominated by wound healing/immune response activity, and cell proliferation activity occurs right up to the late pre-molt stages, alongside muscle tissue modeling activity [24]. They also noted, like us, that very little cell proliferation activity occurs after the molt. One observed difference, however, was the absence of cell proliferation in the very earliest regeneration stages shown in the *P. hawaiensis* study. Our study showed cell cycle activity occurring at its highest levels right from Stage 1. This could potentially be explained by the differing methods of limb ablation. Where the other study cut the limbs in the middle of the segment, ours utilized autotomy, a process that involves considerably less tissue damage. Our process likely required a much shorter healing period, allowing proliferation to begin immediately.

Interestingly, our results differ somewhat from those presented by another study looking at limb regeneration in the crab *Eriocheir sinensis* [25]. A major finding of that work was the significant upregulation of Innexin and other genes enriched for GO terms such as cell migration, response to stimulus, and cell differentiation (Slc7a5, Sox14, Ago2, Fli1, Elf2, Wnt8b, and Vgfr1) in the earliest stage of regeneration, suggesting these are critical to initiating the process. The same genes did not appear in our DEG set, so they were not included in our analysis. A brief investigation found, however, that they were downregulated after the molt as in the *E. sinensis* study and, in some cases, expressed a trend towards upregulation in Stages 2 and 4 like some of the target genes in this paper Other aspects of gene expression in the *E. sinensis* study that did concur with our findings were the upregulation of some muscle synthesis genes in the later pre-molt stage and exoskeletal cuticle genes after the molt.

### 3.2. Target Gene Analysis

#### 3.2.1. Endocrine Factors

Based on previous crustacean studies involving growth factors [19,26,37,38,39], we expected growth factor activity to be present early in the regeneration process and were interested to see where else they might be active. The activity of most of the receptors (BMPR, TGFR, EGFR, FGFR, and EcR) show a two-peak expression pattern, largely in Stages 2 and 4 (Figure 6). Studies on limb regeneration of the fiddler crab *U. pugilator* have shown that the molt hormone receptor EcR is upregulated during two high-growth periods that occur before the molt event [20,26]. Because the putative *C. quadricarinatus* EcR in our data was confidently identified based on close homology and phylogenetic clustering among lobster nuclear receptors (Figure 7), it can be presumed that our two observed peaks likely reflect these other known high-growth phases. Although many of the growth factor ligands and the IR do not follow this two-peak pattern, a number of cell cycle (Figure 9) and pluripotency (Figure 10) target genes do. Taken together, these results suggest that Stage 2 and Stage 4 in our transcriptome are likely key stages for stem cell activity.

Interestingly, there are notable differences between ligand and receptor expression for several growth factors. For BMP2, TGFb, and EGF, the variance could be explained by particular expression level nuances between the pairs, or the presence of alternative ligands and receptors at certain times. For instance, previous studies have shown that circulating ecdysteroid titers do not correspond directly with EcR expression levels during limb regeneration [20]. The contrast between FGF and its receptor, however, is quite stark with completely opposing patterns. This may suggest that, although the identified genes showed the greatest homology with their model organism counterparts, these two homologs may not be a bona fide ligand–receptor pair. Alternatively, after transcription, the ligand may be translated and stored in vesicles for later release, or negative feedback systems may be regulating receptor transcription, as has been shown in other growth factor ligand–receptor pairs [40]. Further investigation to detect protein expression with mass spectrometry or IHC would help to elucidate this.

There were numerous insulin-like growth factor-related genes identified including an IGF/ILP homolog, Cq-ILP1 (AIU40992.1) [41], a known insulin-like growth factor binding protein, Cq-IGFBP (AGS78412.1) [42], several other putative binding proteins (IBPs), and an IR (SF4). A previously identified insulin-like androgenic gland factor, Cq-IAG (ABH07705.1) [43], and Cq-ILP2 (AIU40993.1) [41] could not be located in the current dataset, implying these particular insulin-related genes are not involved in limb regeneration. In concurrence with the DEG analysis, the insulin-related genes show upregulation either at the earlier or the later stages rather than in the two pre-molt growth peaks observed for other target genes. The receptor and two IBPs are expressed earlier, with the former coinciding with just the Stage 2 growth peak, while the other IGFBP is expressed mostly in Stage 5. The IGF/ILP and a single insulin-like growth factor binding protein (SIBD2) are expressed across the whole process though with the highest peaks at Stages 1 and 5. The later stage upregulation likely points to involvement in hypertrophy and related metabolic activity, whereas the earlier upregulation could indicate involvement in cell proliferation and immune response.

A myostatin (MSTN) homolog could not be found in the current dataset, although one has been computationally identified (XP_053657516.1). The closest match was the more confidently identified BMP2 (Table 5). Although there has been some contention around whether MSTN is a negative or positive regulator of crustacean muscle growth, the evidence seems weighted towards the negative role that it has in vertebrates [15,44]. Its absence in the current limb regeneration dataset might support this, implying it may have no role in a situation requiring rapid and extensive muscle growth. However, although MSTN is produced primarily in skeletal muscle in vertebrates, in crustaceans it is also produced in other tissues, including the Y organ [44], suggesting that its involvement cannot be excluded.

#### 3.2.2. Myogenic Factors

In line with the DEG analysis, all myofibrillar proteins (MHC, MLC, MYL, TPM, and ACTA) are significantly upregulated in Stage 6 when the majority of hypertrophy is thought to occur (Figure 8). We were particularly interested in several myogenic transcription factors that could be used as MSC markers and also potentially in molecular experiments to drive myogenic differentiation, which has been conducted previously in CM research [45]. In vertebrates, key myogenic factors are the satellite cell markers, Pax3 and Pax7, and the myogenic regulatory factors (MRFs) MyoD, Myf, Mrf4, and Myogennin [46,47]. In *D. melanogaster*, the main factors are Mef2, Nautilus (Nau), and Twist, where Mef2 is important for fusion, Nau is closest to MyoD in sequence but has only a restricted somatic myogenic function, and Twist is the most likely equivalent of MyoD in terms of embryonic myogenesis [48,49]. Similar relationships between these transcription factors have also been suggested in crustacean species. In *P. hawaiensis* embryonic development, Twist and Mef2 expression have been shown to be somewhat similar to the *D. melanogaster* model with Twist appearing first (although not as early) and seemingly required for Mef2, which again, is required for later muscle determination and differentiation [50]. Like MSTN, a Cq-Twist has been computationally identified (XP_053656043.1) but was not identified in the current dataset, suggesting its embryonic role may not be a feature of limb regeneration. The *D. melanogaster* homologs of Pax3 include Paired and Gooseberry but these share its embryonic patterning role rather than its other vertebrate role as a satellite cell marker [51,52]. Zfh1 appears to be a better candidate in that respect [53,54]. Interestingly, although Pax3 and Gooseberry homologs have been found in the regenerating limbs of other crustaceans [27,55], we were unable to identify them here. It remains to be seen, however, if these or Zfh1 are better markers of crustacean satellite cells.

We expected all of the myogenic transcription factors to be downregulated after the molt, presumably when fully differentiated muscle tissue has replaced differentiating tissues. Only Mef2 showed the expected pattern with upregulation in Stage 4 and very limited expression in Stages 5 and 6. The two potential satellite cell-related genes, Zfh1 and PaxB1, show continuous expression throughout the regeneration process, albeit slightly higher in the earlier stages. This may represent the continued presence of satellite cells in mature muscle tissue. As Twist could not be located in the current transcriptome, we had hoped the other MyoD homolog, Nau, might prove to be a potential myoblast marker; however, it shows most expression after the molt (Figure 8 and Figure 12), indicating it has a role in more developed muscle tissue. In *D. melanogaster*, Nau is actually considered an equivalent of the whole MRF suite rather than just MyoD [48], and as Mrf4 is known to be expressed in mature muscle tissue [56], the expression of Nau here may be more representative of its role than MyoD’s. To our knowledge, apart from a computationally identified myogenic determination protein-like record (XP_053654763.1), no crustacean equivalent of Nau has been characterized. Thus, Mef2 may be a better MSC marker here. Identifying clear markers of proliferating MSCs such as myoblasts is of key importance to CCM development, particularly for stem cell characterization and isolation, so further investigation of these putative myogenic factors is critical.

#### 3.2.3. Cell Cycle and Pluripotency Factors

The RNA-Seq results show that very little PSC activity appears to be present after the molt (Figure 10). This is supported by the qPCR results which show significantly higher expression in Stage 4 than Stage 6, for both Oct3/4 and PCNA (Figure 12). As has been noted, the RNA-Seq results also show that the majority of cell cycle and pluripotency factors follow the same two-peak pre-molt expression pattern as the growth factor receptors. The fact that some pluripotency genes are expressed equally in both stages might suggest one of two things: that these genes are also being expressed by more differentiated stem cells present in more developed tissues (i.e., not true PSCs), or that populations of true PSCs can be accessed at both stages. This does warrant further study; however, in either scenario, both stages should be earmarked as potential sources of proliferative and ‘somewhat’ potent stem cells for CCM development.

### 3.3. Implications

Regenerating limb tissues present a stem cellrich, yet non-lethal alternative to other tissue sources for CCM development. Our study showed that Stage 2 and Stage 4 tissues appear to possess the most stem cell activity in terms of growth, cell cycle, and pluripotency factor gene expression. As Stage 4 tissues are considerably larger, they are likely to be a more practical and prolific source of cells. However, to properly compare the CCM potential of the different stage tissues, their performance in cell culture should be measured. To achieve this, limb regenerates of varying stages can be accessed via autotomy and then subjected to decontamination, digestion, isolation, and culture techniques that are commonly used in primary cell culture. The longevity of cultures and proliferation rates can be compared, as well as the content of particular cell types relevant to CM development. Assessing cell type, however, requires the development of new CM- and crustacean-specific cell characterization tools such as antibodies. These tools must be able to identify stem cell, proliferation, and myogenic markers, and be suitable for applications such as flow cytometry, Western blot, or immunocytochemistry. Factors identified in the present work, along with those already available in public databases, can assist in developing these tools. These can then be utilized to compare the CCM potential of Stage 4 limb regenerates to those of other stages, and also to other crustacean tissues.

### 3.4. Limitations

Our results have highlighted some limitations in the experimental process. Most notably, as shown in the PCA in Figure 1, there is considerable cross-over in gene expression among the earlier stages which suggests our approach to stage allocation might not have been the most appropriate. Rather than trying to group limbs into a minimal number of stages based mostly on morphology, a more uniform sampling of R-values may have been better. Regeneration, like all biological processes, is graduated; there may be peaks and troughs, but it is not conducted in succinct steps. Thus, expecting to see uniformity within our designated stages might be unrealistic. While two different limb buds may look morphologically similar, molecular activity (mRNA levels) within could be vastly different, particularly due to the molt stage. Although extremely important to the regeneration process, the molt stage could not be accounted for in the earlier stages, as this is extremely difficult in our species without dissection.

Another limitation was the absence of pre-regeneration tissues in the RNA-Seq samples. The Stage 6 samples, which were considered representative of intermolt, non-regenerating tissues, were taken 2–3 weeks after the molt and may still have been undergoing some level of regeneration. In contrast, the Stage 6 tissues for qPCR analysis were taken from separate animals that had not undergone recent limb regeneration. RNA taken from fully grown limbs prior to autotomy may have better represented non-regenerating tissues for both the RNA-Seq and qPCR work.

## 4. Materials and Methods

### 4.1. Animal Handling

Twenty-four adult female Australian Redclaw crayfish, *Cherax quadricarinatus*, were purchased from Freshwater Australian Crayfish Traders, Tarome, Australia. Crayfish ranged from 30 to 60 gm in weight. They were maintained at the University of the Sunshine Coast aquaculture facility with constant freshwater exchange and fed thrice weekly with commercial fish food pellets (Aquamunch—African Attack) purchased from Aquaholics Aquarium Supplies, Forest Glen, Australia. Crayfish were kept at ambient indoor temperature (22–24 °C) during the months of November through to May; water temperature was not measured. Autotomy was encouraged through the simultaneous application of pressure to the propodus and dactyl on each appendage, allowing the crayfish to tail-flip and swim backwards, releasing the claw in the process. Coxa segments were observed in each individual to be clean and sealed within seconds.

### 4.2. Regeneration Assessment

Over seven months, the regeneration of the claws was photographed every four to ten days and measured with ImageJ software (version 1.53h) Measurement data were used to calculate the R-value, a measure of allometric growth found by dividing the length of the limb by the width of the carapace and multiplying by 100 [19,20]. Based on morphological observations, the regrowing claws of the 24 individuals were identified as being at one of six regeneration stages and selected for surgical removal or re-autotomy (four for each stage). Due to varying molt stages, ages, and health statuses (all unknown), the regeneration progression of each crayfish varied considerably. We observed that the time taken to reach Stage 2 ranged from 26 to 60 days, and the time to molt ranged from 42 to 138 days.

### 4.3. Anesthetic

To anesthetize crayfish, temperature cooling was used in conjunction with the anesthetic/analgesic product Aqui-S, purchased from Primo Aquaculture, Narangba, Australia. This approach was based on previous studies investigating crustacean responses to analgesia [12,57]. Anesthesia was considered effective when crayfish remained motionless despite pressure being applied to their highly innervated anal region. The sufficient concentration was 500 ppm or 915 µL/L of Aqui-S with the optimal temperature being between 5 and 14 °C, and the time taken for crayfish to reach complete anesthesia varied from 14 to 28 min.

### 4.4. Sample Preparation

One tissue sample (0.1–1.0 g) was taken from each of the 24 individuals. Regrowing limb tissues at Stages 1 to 4 were re-automized or, under anesthesia described above, removed with sterile surgical scissors, placed into sterile, RNAse-free Eppendorf tubes, and then snap-frozen in liquid nitrogen before being stored at −80 °C. Stage 5 and 6 limbs were re-autotomized with tissues immediately extracted from the cuticle, collected, frozen, and stored as above. Surgery sites were treated with alcohol wipes and sealed with Liquid Skin^®®^ purchased from Chemist Warehouse, Maroochydore, Australia. Crayfish were monitored in recovery tanks until fully recovered and placed back into their home compartments. Once regrowth of limbs was observed (between one and five weeks), crayfish were placed back into the general population.

### 4.5. RNA Extraction, Quantification, and Sequencing

RNA was extracted from the 24 tissue samples using RNAzol ^®®^ RT (Molecular Research Center, Melbourne, Australia) as previously described [29]. The quality and quantity of all samples were confirmed with NanoDrop 2000 (ThermoFisher, Scoresby, Australia). All samples measured at least 4 µg of RNA. The samples were then stored in RNAse-free Eppendorf tubes at −80 °C until being sent in dry ice to Novogene (Singapore) for quality control using Bioanalyzer, library preparation with TruSeq, and RNA-Seq with the Illumina HiSeq2500 platform with paired-end 150 bp sequencing. Two samples (one from Stage 1 and one from Stage 6) failed quality control (QC) and the remaining twenty-two were sequenced, adaptor-trimmed, and returned as a minimum of 39 million cleaned reads each. All raw read FASTQ files were uploaded to NCBI SRA (https://www.ncbi.nlm.nih.gov/sra) (Accessed 27 June 2024) under BioProject number PRJNA780617.

### 4.6. Transcriptome Assembly and Quantification

Based on QC read reports from Novogene (Singapore), all reads were further trimmed of 5 nucleotides from the 5′ end only, using BBduk (version 38.90) (Trimmed reads from six samples (one from each stage) were assembled de novo using Trinity (version 2.12.0, Broad Institute, Cambridge, MA, USA) [58] with default paired read parameters except for strand-specific settings (RF) and minimum contig length (400 bp), resulting in an initial assembly FASTA file, containing 144,049 contigs. Evidential Gene’s tr2aacds tool (version 2023jul15, Indiana University, IN, USA) was then used to remove redundancies, resulting in a final reference assembly of 55,018 contigs (SF1). BUSCO (version 5.4.3, Swiss Institute of Bioinformatics, University of Geneva, Switzerland) [59] and rnaQUAST (version 2.2.3, Saint Petersburg State University, Russia) [60] were also used to obtain metrics for assessing assembly quality.

Before performing quantification, raw reads were trimmed using fastp (version 0.23.4, HaploX Biotechnology, Shenzhen Institutes of Advanced Technology, Chinese Academy of Sciences, Shenzhen, China) [61] by trimming 15 nucleotides from the 5′ end. SortMeRNA (version 4.3.5, Université de Lille France) [62] was used to remove rRNA from bacteria, archaea, and crustaceans. This involved using rRNA databases that are provided by SortMeRNA (silva-arc-16s-id95, silva-arc-23s-id98, and silva-bac-16s-id90), as well as a custom database which contained deduplicated crustacean rRNA sequences from RefSeq (https://www.ncbi.nlm.nih.gov/refseq/) (Accessed 27 January 2024). Salmon (version 1.10.1, University of Maryland, College Park, MD, USA) [63] was used for quantification (SF2).

### 4.7. Data Upload into CrustyBase.org

The final reference assembly was initially imported into CLC Genomics Workbench 8.0.3 (Qiagen, Clayton, Australia) for quantification. The 22 trimmed read files were then quantified relative to the library size (calculated as reads per kilobase per million reads; RPKM) by mapping them to the imported reference transcriptome with CLC’s RNA-Seq analysis tool, using default parameters. The RNA-Seq results and the final reference assembly were uploaded to the public database CrustyBase.org (http://crustybase.org) (Accessed 21 September 2023) [64]. For CrustyBase to calculate the standard error, all features (stages) must contain the same number of replicates so only three replicates of each stage were uploaded. In this instance, S2.2, S3.4, S4.2, and S5.2 were removed in order for Stages 2, 3, 4, and 5 to have equal replicates (*n* = 3) with Stages 1 and 6. Uploading the data into this public platform provided a second BLAST-searchable database and allowed for easy visualization of domain prediction and contig expression across the six stages.

### 4.8. Differential Expression Analysis

Python (version 3.9.18) was used to generate transcript-to-gene mappings based on the transcript IDs outputted by Evidential Gene. Transcript level abundance was aggregated to gene level abundance with tximport (version 1.30.0, The University of North Carolina at Chapel Hill, NC, USA) [65] (SF3). DESeq2 (version 1.42.0, The University of North Carolina at Chapel Hill, NC, USA) [66] was used for performing differential expression analysis for all pairwise comparisons. This involved using the median of ratios method to estimate size factors for each gene. These were used to normalize counts to account for differences in sequencing depth and library composition across samples. Next, gene-wise dispersion estimates were obtained and shrunk towards a mean-dispersion trend line. Then, count data were modeled using a negative binomial distribution, and the Wald test was applied to identify DEGs. Log2 fold changes (LFC) were shrunk with the ‘ashr’ method [67]. Differentially expressed genes (DEGs) were those with p-adjusted values < 0.01 (adjusted using the Benjamini-Hochberg method), an absolute LFC ≥ 1, and an average expression level ≥ 20. The PCA plot was created after applying the regularized log transformation to counts in DESeq2. The statistical analysis results can be found in SF5.

### 4.9. Reference Assembly Annotation

Annotation of the reference assembly was conducted with the TransPi pipeline (version 1.3.0-rc, Ludwig-Maximilians-University, Germany) using default parameters [68]. This yielded annotations from the SwissProt database (accessed 1 May 2023) as well as information on Gene Ontology (GO), Evolutionary Genealogy of Genes: Non-supervised Orthologous Groups (eggNOG), Kyoto Encyclopedia of Genes and Genomes (KEGG) pathways, and protein family (PFAM) domains (SF4). Heatmaps of all annotated DEGs in all pair-wise comparisons were created with pheatmap (version 1.0.12) in R (version 4.3.2). An additional heatmap was created of the top 10 (greatest fold change) up- and downregulated DEGs from each comparison. In this heatmap, duplicates (contigs with the same SwissProt annotation) were condensed into one record by averaging their normalized counts from DESeq2.

### 4.10. Gene Ontology Enrichment Analysis

GO enrichment analysis was performed by determining whether the number of DEGs in a gene set (GO term) exceeds what would be expected based on the background gene set. GO terms for each gene were identified by TransPi. The background gene set included all genes from the transcriptome. For each pairwise comparison, DEGs were separated into lists of up- and downregulated genes. Then, the topGO package (version 2.54.0, Max-Planck-Institute for Informatics, Germany) [69] was utilized with a nodeSize of 70 to filter out GO terms with fewer than 70 annotated genes, thereby keeping more broad/general terms. Fisher’s exact test (*p* < 0.01) was used to identify GO terms that were significantly overrepresented among the DEGs compared to the background. The default “weight01” algorithm was used as it considers the topology of the GO graph when computing the significance of a GO term.

Summary statistics (number of unique GO IDs in each GO category) were calculated in Python (version 3.9.18) (SF6). From the 227 enriched BP GO terms among each pair-wise comparison, a list of 53 unique terms was created by selecting the one containing the highest number of significant DEGs and then removing terms unrelated to crustacean limbs (e.g., human health-specific) and generic (uninformative) cellular process terms. For visualization using a heatmap, the normalized counts of the contributing DEGs for each GO term were obtained from DESeq2. For each DEG, expression was averaged across replicates in each stage. Then, the expression from all contributing DEGs for that term was summed. Visualization was carried out with ComplexHeatmap (version 2.20.0, German Cancer Research Center, Germany) [70].

### 4.11. Graphical Representation of the Functional DEG and GO Term Categories

After the top 92 DEGs and enriched GO terms were categorized into 14 categories (Table 4, Section 2.7), the average gene count value for each category was parsed in Python (version 3.8) to average per stage data and convert the absolute expression values into log-scaled columns to compare absolute expression between the graphs using the matplotlib library matplotlib [71].

### 4.12. Target Gene Analysis

UniProt and Flybase (http://www.flybase.org) (Accessed 21 June 2024) amino acid sequences of well-annotated *D. melanogaster*, *Mus musculus*, and *Homo sapiens* proteins were selected for a tBLASTn search against the transcriptome in CLC and CrustyBase. The top hits for each homolog were annotated with NCBI BLAST search and domain prediction with SMART and CrustyBase and selected based on the combined lowest E-value, highest expression values, and the presence of at least one predicted domain. Normalized gene count data were parsed in Python (version 3.8) and plotted as for the DEG and GO term function categories.

### 4.13. Phylogeny

Phylogeny was conducted to compare the Redclaw EcR with *P. ornatus* nuclear receptors (Figure 7). Amino acid sequences were aligned by ClustalW [72] in MEGA (version 10.1.7) [30] with default parameters and then subject to phylogenetic analysis using the Neighbor-Joining method with a p-distance model, and a bootstrap consensus tree was inferred from 1000 replicates.

### 4.14. Histology

Tissue samples from Stage 4 and Stage 6 claws were collected, fixed, sectioned, and stained for histological analysis as previously detailed [73]. Briefly, tissues were fixed in Bouin’s Fixative Solution (Sigma, Melbourne, Australia), dehydrated in sequentially concentrated ethanol (50–100%), followed by two washes in xylene, embedded in paraffin, and sectioned into 5 µm slices. The slides were then deparaffinized in xylene and rehydrated in sequentially reduced ethanol concentrations (100–50%), followed by deionized water. One slide of each sample stage was stained with hematoxylin (4 min) and eosin (3 min), then dehydrated in ethanol and xylene again, air-dried in a fume hood, and sealed with xylene glue and a coverslip before being photographed with a compound light microscope. Additional slides of each sample stage after rehydration were washed in PBS (2 × 5 min), stained with DAPI nuclear stain, sealed with a coverslip, and then photographed with a confocal microscope (Figure 11).

### 4.15. Real-Time Quantitative PCR

RNA was extracted from new Stage 4 and Stage 6 samples (six samples/animals per stage) as outlined in Section 4.4 and Section 4.5 above. Concentration and quality were checked with Nanodrop 2000 (ThermoFisher, Scoresby, Australia), and 1 µg samples were used to synthesize cDNA with the Tetro cDNA kit (Bioline, Frenchs Forest, Australia) as detailed previously [74]. Briefly, each 1 µg RNA sample was used to synthesize cDNA mixed with 1 µL dNTPs and 1 µL random hexamers before being heated at 70 °C for 5 min. Samples were then mixed with a 4:1:1 ratio of kit-supplied buffer, RNAse inhibitor, and Reverse Transcriptase and heated at 45 °C for 45 min, then 85 °C for five minutes, and checked for cDNA quality and concentration before being stored at −20 °C.

For qPCR, primers appropriate for the Universal Probe Library (previously supplied by Roche, no longer available) for select target genes were designed using the Neoform primer tool (https://primers.neoformit.com/) (Accessed 5 May 2023) (Table 6). Cq-18S primers were used as a positive control and to normalize the relative expression of the target genes. cDNA samples for target genes were mixed with a 1:1:0.05:5:9 ratio of forward primer, reverse primer, Roche Universal Probe, FastStart Universal Probe Master (Rox) (Sigma/Merck, Sydney, Australia), and ultra-pure water and run in a Rotor-Gene Q qPCR machine (, Qiagen, Clayton, Australia) for 45 cycles. cDNA samples for the 18S reactions were diluted 1:100 with ultra-pure water.

The relative expression was calculated as described previously [75] using the formula 2^−ΔΔ^CT, where CT was the cycle threshold before amplicon concentration fluorescence plateaued. 18S expression was used to normalize and calculate the relative expression values for the gene targets. The mean values ± standard error of the mean (SEM) were subjected to a Kruskal–Wallis test to identify significant differences between Stages 4 and 6 for all target genes (*p* < 0.05, Test Statistic = 21.043) (IBM SPSS Statics, version 29).

## 5. Conclusions

Regenerating crustacean limbs present a stem cell-rich and non-lethal source of tissue for CCM development. As the first to do so in this context, we investigated stem cell gene expression across the limb regeneration process in the crayfish *C. quadricarinatus*. Our RNA-Seq, histological, and qPCR results suggest that pre-molt limb regeneration tissues harbor more stem cell activity than post-molt or fully developed limb tissue. Two pre-molt stages (Stages 2 and 4) displayed increased activity for growth factor receptors (EGFR, TGFR, BMPR, and FGFR), cell cycle genes (Cyclins, CDKs, p53/63, PCNA, and Retinoblastoma), and pluripotency factors (C-Myc, Klf4, Sox2, Wnt2/4, Hedgehog, and Frizzled). As Stage 4 limbs are larger, they likely offer a more practical tissue source for cell culture experiments. We have identified ways these and other tissues can be further evaluated with cell culture experiments and recommend the development of new tools to support this. With a specific focus on non-lethality, this work supports the advancement of the CCM field and more sustainable seafood production.

## Figures and Tables

**Figure 1 ijms-25-08623-f001:**
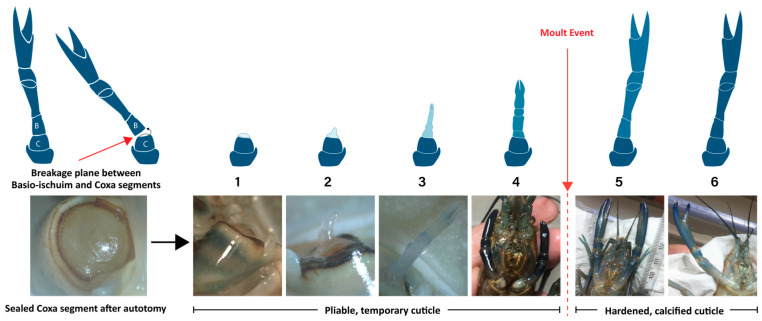
Graphical depiction and photographs of six stages of *Cherax quadricarinatus* claw regeneration in relation to molt cycle. Allocation of stage was based on observed morphology (shape, segmentation, allometric growth, and pigment). Source: [15].

**Figure 2 ijms-25-08623-f002:**
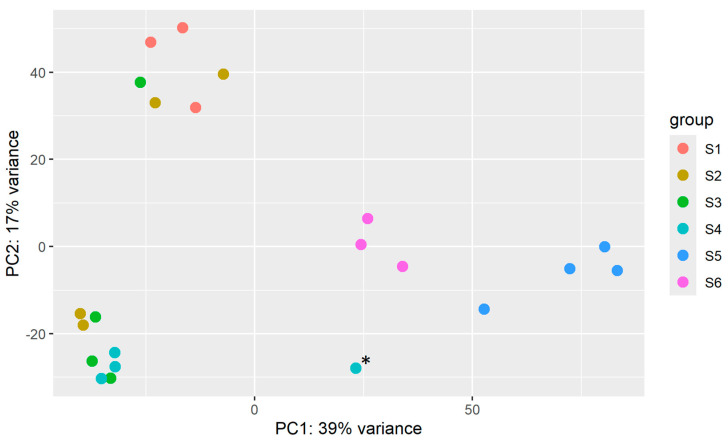
Principal component analysis of the top 500 genes with the highest row variance across six stages of crayfish claw regeneration. The asterisk (*) denotes a single outlier in the Stage 4 sample identified as S4.2 which has a notably larger R-value than the other three samples (Table 1).

**Figure 3 ijms-25-08623-f003:**
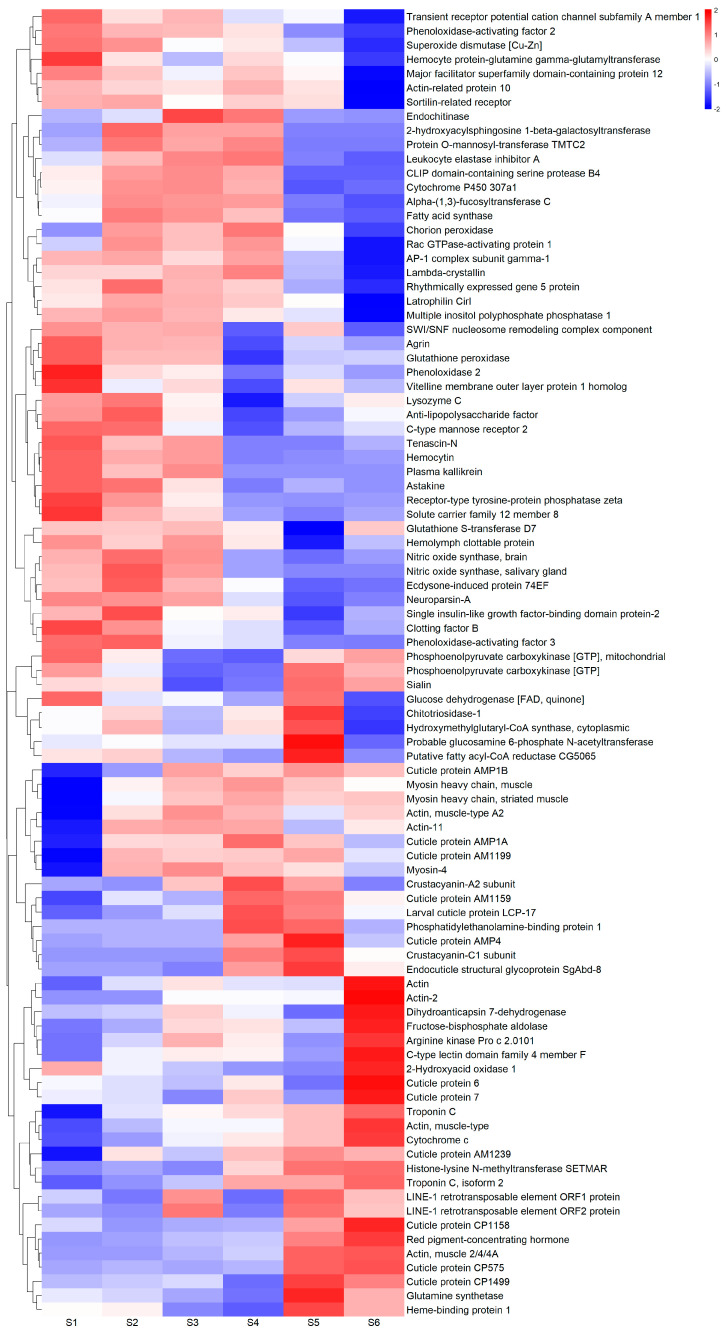
A heat map of the top 10 up- and down-regulated differentially expressed genes (DEGs) with SwissProt annotations within all 15 pair-wise comparisons between six different stages of crayfish claw regeneration (S1 to S6 on the *X* axis). The scale bar represents log2 (normalized counts + 1).

**Figure 4 ijms-25-08623-f004:**
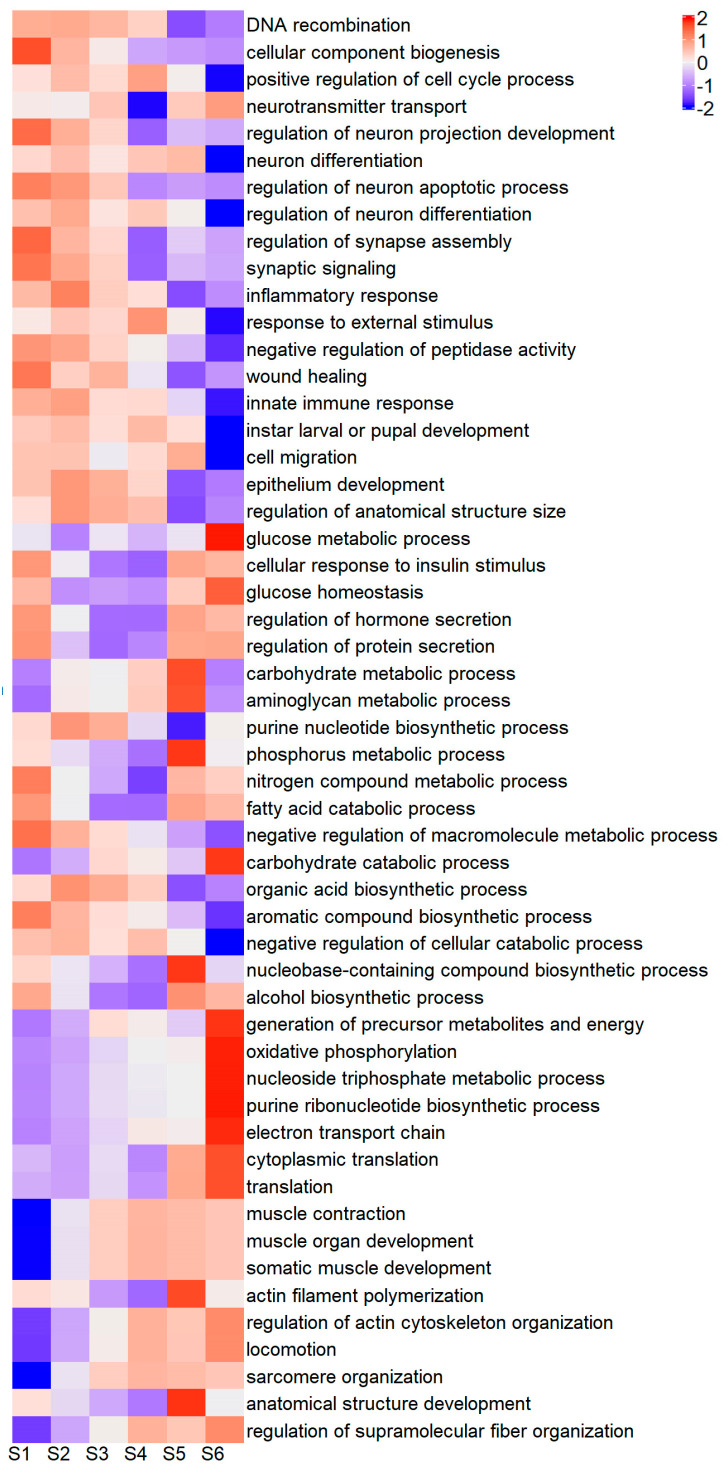
Heatmap of normalized gene counts of all differentially expressed genes (DEGs) contributing to unique enriched GO terms. Scale bar represents log2 (normalized counts + 1).

**Figure 5 ijms-25-08623-f005:**
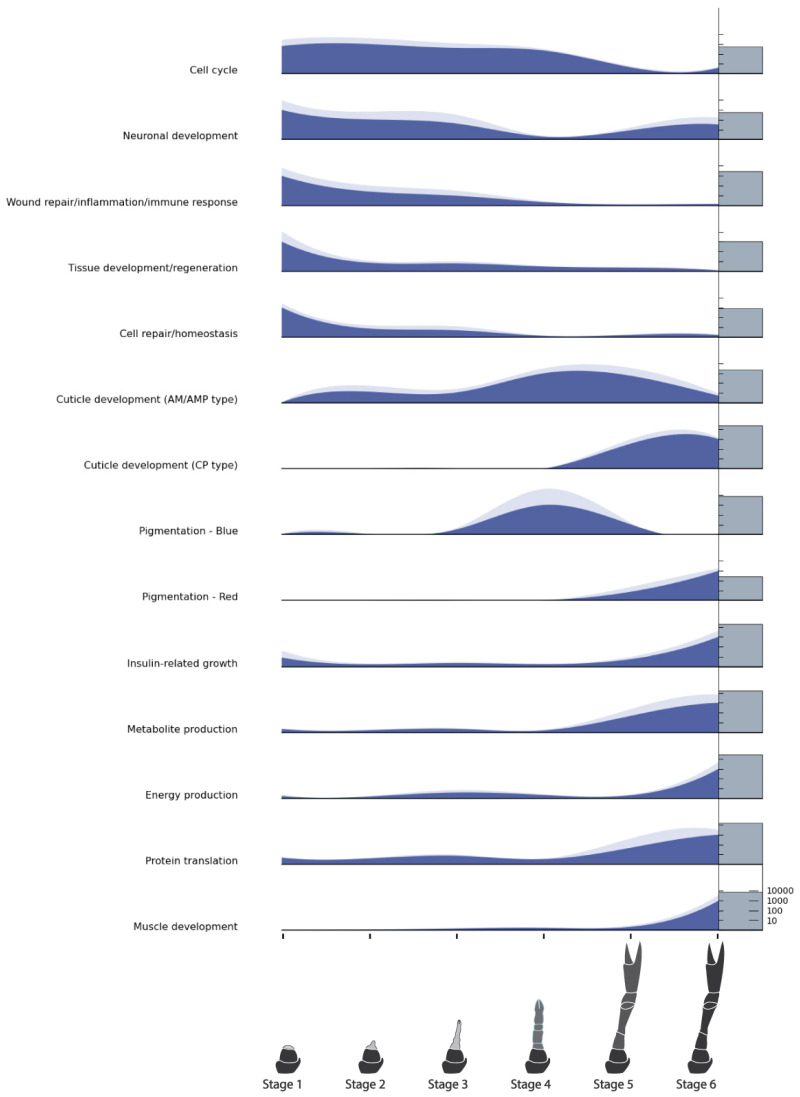
Relative expression of differentially expressed genes (DEGs) and enriched GO terms across six limb regeneration stages in the crayfish *Cherax quadricarinatus* grouped into 14 functional categories. Each level of the plot represents the average relative expression of genes in each category, measured as the mean normalized gene count (dark blue area) for each stage, with standard error (pale blue area). The absolute gene count is shown as a log-scaled gray column on the right-hand edge of each expression plot; the first level includes a scale bar that applies to all levels.

**Figure 6 ijms-25-08623-f006:**
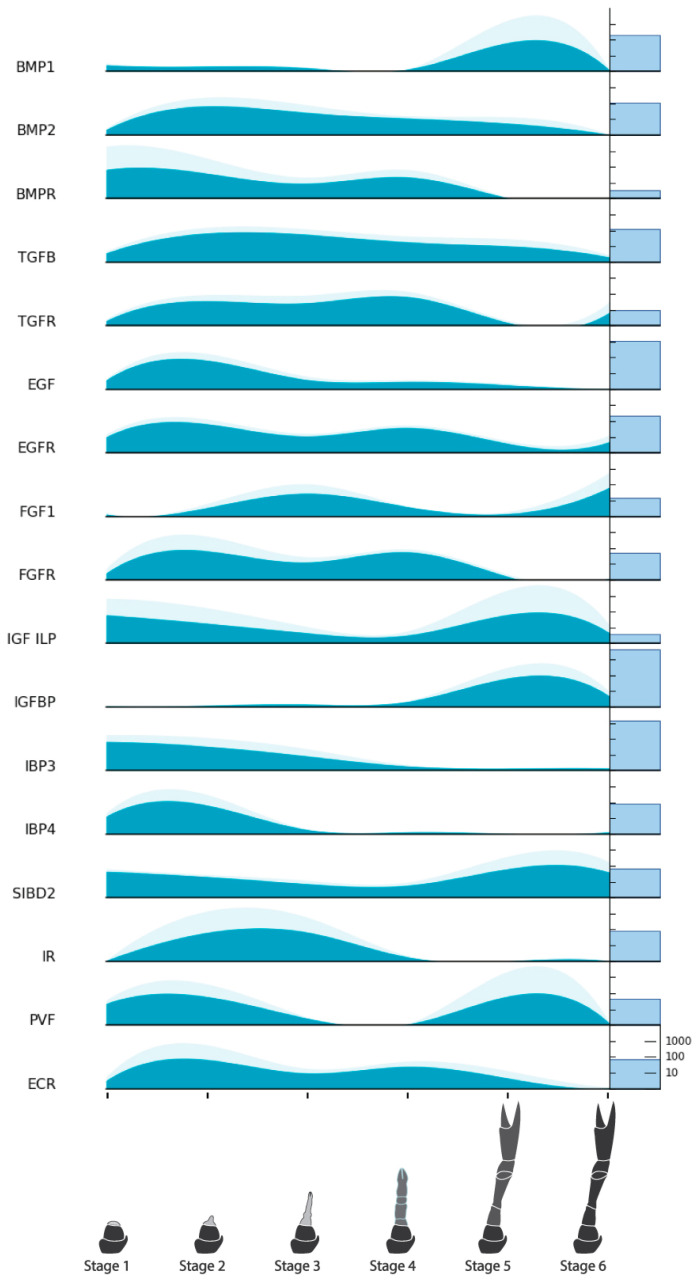
Relative expression of endocrine factors across limb regeneration of *Cherax quadricarinatus*. Mean normalized gene count, standard error, and absolute gene counts calculated and represented as in Figure 5. Gene names are abbreviated. For full gene names, refer to Table 5 and section above.

**Figure 7 ijms-25-08623-f007:**
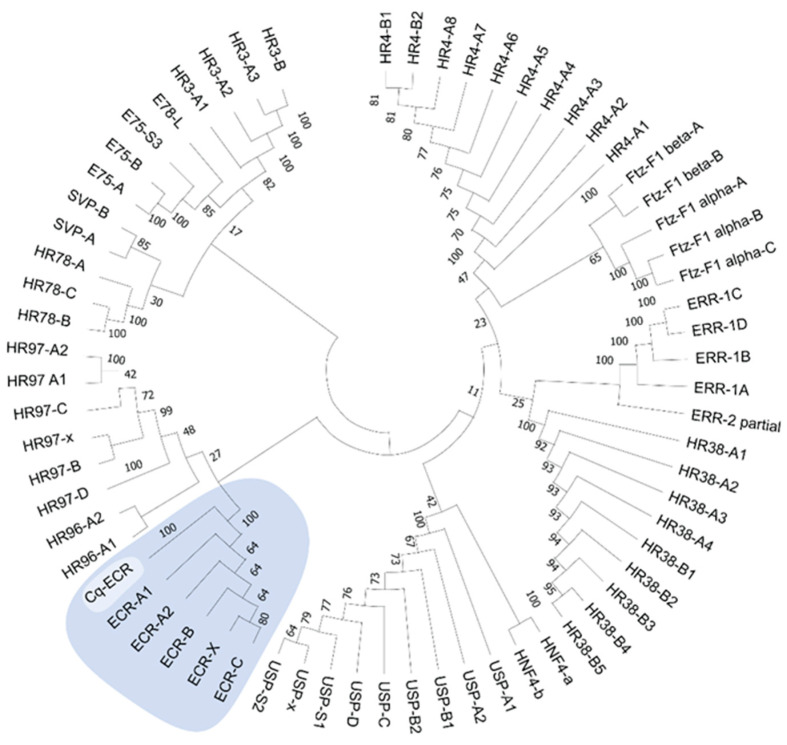
A phylogenetic analysis showing *Cherax quadricarinatus* ecdysone receptor (Cq-ECR) clusters with ECRs amongst several *Panulirus ornatus* nuclear receptor families. *P. ornatus* nuclear receptor sequences were obtained from publicly available supplementary data [29]. The analyses were conducted with the Neighbor-Joining method using the p-distance model, and the bootstrap consensus tree was inferred from 1000 replicates. Bootstrap percentage values are shown at each node. Evolutionary analyses were conducted in MEGA X 10.1.7 [30].

**Figure 8 ijms-25-08623-f008:**
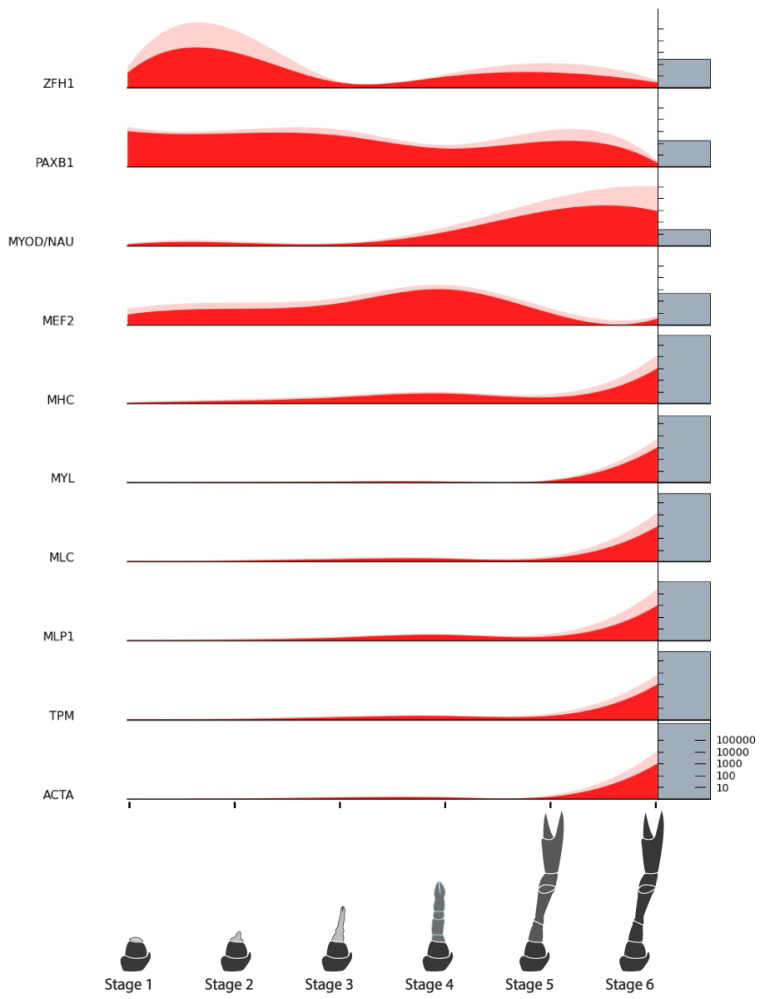
Relative expression of myogenic factors across limb regeneration of *Cherax quadricarinatus*. Mean normalized gene count, standard error, and absolute gene counts calculated and represented as in Figure 5. Gene names are abbreviated. For full gene names, refer to Table 5 and section above.

**Figure 9 ijms-25-08623-f009:**
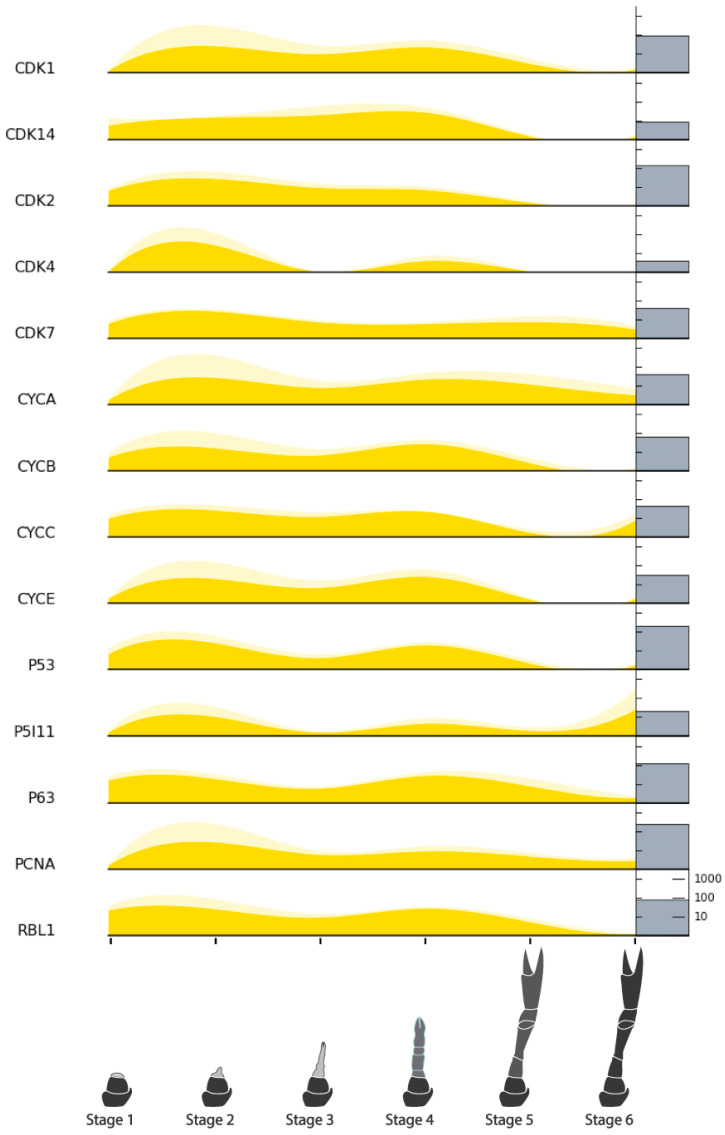
Relative expression of cell cycle factors across limb regeneration of *Cherax quadricarinatus*. Mean normalized gene count, standard error, and absolute gene counts calculated and represented as in Figure 5. Gene names are abbreviated. For full gene names, refer to Table 5 and section above.

**Figure 10 ijms-25-08623-f010:**
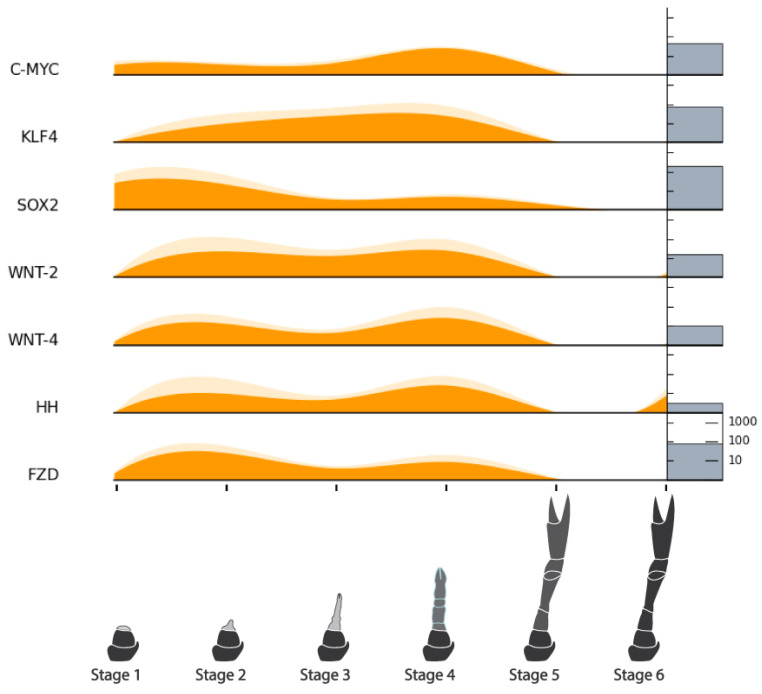
Relative expression of pluripotency factors across limb regeneration of *Cherax quadricarinatus*. Mean normalized gene count, standard error, and absolute gene counts calculated and represented as in Figure 5. Gene names are abbreviated. For full gene names, refer to Table 5 and section above.

**Figure 11 ijms-25-08623-f011:**
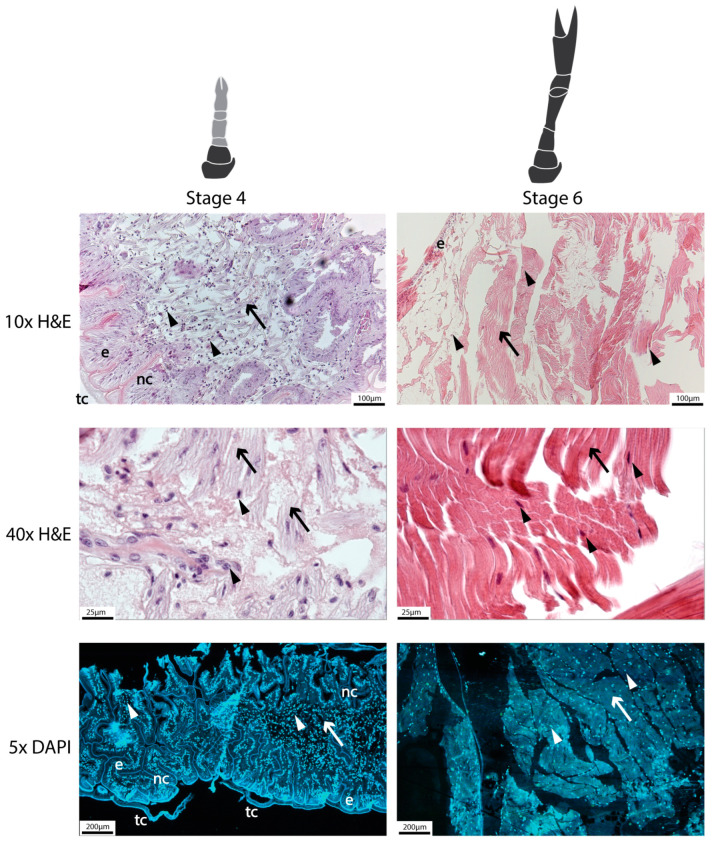
Histological analysis showing greater ratio of nuclei to fiber/tissue in Stage 4 vs. Stage 6 regenerating limbs. Hematoxylin and eosin (H&E) images are at 10× and 40× magnification and DAPI staining is at 5× magnification. Notations include nuclei (arrowheads), muscle fibers (arrows), epithelial tissue (e), temporary cuticle (tc), and new cuticle (nc).

**Figure 12 ijms-25-08623-f012:**
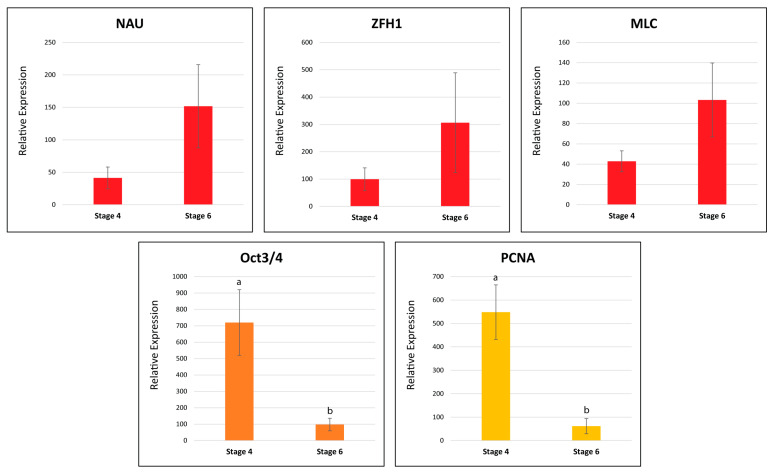
Real-time quantitative PCR results showing relative expression between Stage 4 and Stage 6 tissues for five tested target genes: three myogenic factors (red) Nau, MLC, and Zfh1, pluripotency factor (orange) Oct3/4, and cell cycle factor (yellow) PCNA. Data represent mean +/− SEM, *n* = 6. Letters (a, b) denote significant differences in expression after pair-wise comparison using Kruskal–Wallis test (*p* < 0.05).

**Table 1 ijms-25-08623-t001:** Twenty-four tissue samples categorized into six stages of claw regeneration showing R-value (Rv) which is a measure of allometric growth (regenerating limb length divided by carapace length × 100) to account for whole-body size differences among individuals. Rv for Stage 6 individuals was not measured. Stages 5 and 6 include number of days post molt (dpm).

Stage	Sample	R-Value *	Avg Rv	Stage	Sample	R-Value *	Avg Rv
Stage 1	S1.1	1.21	0.89	Stage 4	S4.1	29.95	49.49
	S1.2	1.17			S4.2	68.31	
	* S1.3 *	* 0.23 *			S4.3	36.24	
	S1.4	0.97			S4.4	26.45	
Stage 2	S2.1	2.37	3.41	Stage 5	S5.1	138.59	136.36 (1–3 dpm)
	S2.2	4.81			S5.2	145.60	
	S2.3	3.62			S5.3	158.20	
	S2.4	2.85			S5.4	103.05	
Stage 3	S3.1	24.83	22.85	Stage 6	S6.1	NA	NA (18–21 dpm)
	S3.2	20.93			S6.2	NA	
	S3.3	21.47			* S6.3 *	* NA *	
	S3.4	24.19			S6.4	NA	

* R-value at time of tissue extraction; Avg Rv = average R-value of four samples. Italicized red text denotes two RNA samples that failed initial quality control and were not included in subsequent analyses.

**Table 2 ijms-25-08623-t002:** Final reference assembly quality statistics after redundancy reduction.

Complete	85.1%
Single and complete BUSCO	44.1%
Duplicated and complete BUSCO	41.0%
Fragmented BUSCO	6.9%
Missing BUSCO	8.0%
Transcripts	55,018
Transcripts > 500 bp	44,584
Transcripts > 100 bp	25,516
Average length of assembled transcripts	1488.461
Longest transcript	22,480
Total length	81,892,142
Transcript N50	2242

**Table 3 ijms-25-08623-t003:** Summary of reference assembly annotation.

Number of transcripts	55,018
Total number of ORFs	43,515
ORFs with annotations	39,360
ORFs without annotations	4155
ORFs complete	17,773
ORFs complete with annotations	16,223
Transcripts with eggNOG COG	79
Transcripts with GO	23,105
Transcripts with KEGG	18,749
Transcripts with PFAM	23,635
Transcripts with BLAST	21,444

**Table 4 ijms-25-08623-t004:** Functional categorization of the top differentially expressed genes (DEGs) and those contributing to the top enriched GO terms, resulting in 14 combined categories.

DEG Categories	GO Term Categories	Combined Categories
Neuronal development	Cell cycle	Cell cycle
Wound repair/inflammation/immune response	Neuronal development	Neuronal development
Tissue development/regeneration	Wound repair/inflammation/immune response	Wound repair/inflammation/immune response
Cell repair/homeostasis	Tissue development/regeneration	Tissue development/regeneration
Cuticle development (AM/AMP type)	Insulin-regulated growth	Cell repair/homeostasis
Cuticle development (CP type)	Metabolite production	Cuticle development (AM/AMP type)
Pigmentation—blue	Energy production	Cuticle development (CP type)
Pigmentation—red	Protein translation	Pigmentation—blue
Metabolite production	Muscle development	Pigmentation—red
Energy production		Insulin-regulated growth
Muscle development		Metabolite production
		Energy production
		Protein translation
		Muscle development

**Table 5 ijms-25-08623-t005:** Endocrine, myogenic, cell cycle, and pluripotency factors relevant to stem cell proliferation and differentiation of muscle tissue growth, potentially present during crustacean limb regeneration [15]. Those with transcript IDs were identified in the current dataset. Refer to figures noted for relative expression of each target factor.

Gene Category	Gene/Factor	Transcript_ID	Expression
Endocrine Factors	Bone Morphogenetic Protein 1 (BMP1)	NonamEVm007201t1	Figure 6
	Bone Morphogenetic Protein 2 (BMP2)	NonamEVm003174t1	
	Bone Morphogenetic Protein Receptor (BMPR)	NonamEVm006724t1	
	Transforming Growth Factor Beta (TGFβ)	NonamEVm004397t1	
	Transforming Growth Factor Beta (TGFβR)	NonamEVm003982t1	
	Ecdysone Receptor (ECR)	NonamEVm002506t1	
	Epidermal Growth Factor (EGF)	NonamEVm000011t1	
	Epidermal Growth Factor Receptor (EGFR)	NonamEVm000307t1	
	Fibroblast Growth Factor 1 (FGF1)	NonamEVm009793t1	
	Fibroblast Growth Factor Receptor (FGFR)	NonamEVm002596t1	
	Insulin-like Growth Factor/Peptide (IGF/ILP)	NonamEVm015416t1	
	Insulin-like Growth Factor Binding Protein (IGFBP)	NonamEVm006292t1	
	Insulin-like Growth Factor Binding Protein 3 (IBP3) Insulin-like Growth Factor Binding Protein 4 (IBP4) Insulin-like Growth Factor 2 mRNA-Binding Protein (IGF2B)	NonamEVm007418t1 NonamEVm018337t1 NonamEVm002726t1	
	Single Insulin-like Growth Factor-Binding Domain Protein-2 (SBD2)	NonamEVm018319t1	
	Insulin Receptor (IR)	NonamEVm000244t1	
	Platelet-Derived/Vascular Endothelial Factor (PVF)	NonamEVm006397t1	
	Platelet-Derived/Vascular Endothelial Factor Receptor (PVR)	Not found	
	Hepatocyte Growth Factor and Receptor (HGF/HGFR)	Not found	
	Myostatin (MSTN)	Not found	
Myogenic Factors	Zinc Finger Homeodomain 1 (ZFH1)	NonamEVm000901t1	Figure 8
	Pax 3 and Pax 7 Binding Protein 1 (PAXB1)	NonamEVm001208t1	
	Myogenic Determination Protein/Nautilus (MYOD/NAU)	NonamEVm006620t1	
	Myocyte Enhancer Factor 2 (MEF2)	NonamEVm004258t1	
	Myosin Heavy Chain, Muscle (MHC)	NonamEVm000136t1	
	Myosin Regulatory Light Chain (MYL)	NonamEVm011187t1	
	Myosin Light Chain (MLC)	NonamEVm013270t1	
	Muscle Lim Protein (MLP)	NonamEVm002170t1	
	Tropomyosin (TMP)	NonamEVm005681t1	
	Actin, Muscle-Related (ACTA)	NonamEVm004870t1	
	Myogenic Determination Protein/Twist (MYOD/TWIST)	Not found	
	Paired Box Protein 3 (PAX3)	Not found	
	Paired Box Protein 7 (PAX7)	Not found	
Cell Cycle Factors	Cyclin-dependent Kinase 1 (CDK1)	NonamEVm006446t1	Figure 9
	Cyclin-dependent Kinase 14 (CDK14)	NonamEVm007728t1	
	Cyclin-dependent Kinase 2 (CDK2)	NonamEVm006000t1	
	Cyclin-dependent Kinase 4 (CDK4)	NonamEVm010173t1	
	Cyclin-dependent Kinase 7 (CDK7)	NonamEVm005270t1	
	Cyclin A (CYC-A)	NonamEVm003889t1	
	Cyclin B (CYC-B)	NonamEVm004508t1	
	Cyclin C (CYC-C)	NonamEVm007180t1	
	Cyclin E (CYC-E)	NonamEVm006398t1	
	Cellular Tumor Antigen p53 (P53)	NonamEVm004102t1	
	Tumor Protein p53-inducible Protein 11 (P5I11)	NonamEVm007863t1	
	Cellular Tumor Antigen p53 (P63)	NonamEVm001352t1	
	Proliferating Cell Nuclear Antigen (PCNA)	NonamEVm007483t1	
	Retinoblastoma-like Associated Protein (RBL1)	NonamEVm000671t1	
	Cyclin-dependent Kinase 3 (CDK3)	Not found	
	Cyclin-dependent Kinase 6 (CDK6)	Not found	
	Cyclin D (CYC-D)	Not found	
Pluripotency Factors	MYC Proto-oncogene Protein (C-MYC)	NonamEVm003188t1	Figure 10
	Krueppel-like Factor 4 (KLF4)	NonamEVm004452t1	
	SRY-Box Transcription Factor 2 (SOX2)	NonamEVm004482t1	
	POU Domain, Class 5, Transcription Factor 1 (OCT3/4)	NonamEVm004528t1	
	Int/Wingless Family Protein 2 (WNT-2)	NonamEVm005041t1	
	Int/Wingless Family Protein 4 (WNT-4)	NonamEVm003744t1	
	Hedgehog (HH)	NonamEVm003556t1	
	Frizzled (FZD)	NonamEVm005809t1	
	Nanog (NANOG)	Not found	
	Int/Wingless Family Protein 1 (WNT-1)	Not found	

**Table 6 ijms-25-08623-t006:** Primers designed for selected target genes and Roche Universal Probes (discontinued). The 18S housekeeping gene was used for the positive control and to normalize the expression of the other genes.

Primer Name	UPL Probe #	Forward Primer	Reverse Primer
qCq18S	133	GCGCTACACTGAAGGGATCA	AGGGGTTTGAACGGGTTACC
qCqMLC	135	CGTGTGCTGGGAATCACTGA	CGGGCTGGACCTTCGTTATT
qCqZfh1	10	AGAAGTCCCCCTCATCTCCC	CAACGCAACTATTAGCCCGC
qCqPCNA	42	GGCTCGTCTTGTCCAAGGAA	TGACGCTTCATTCAGCAGCT
qCqOct3/4	85	CGAAGTCGAAGAGGAGACCC	CCAGCTTGATACGCCTCTGT
cCqNautilus	5	CGTGCAAGAGAAAGAGCGTC	TCACCTTCCGTAGTCTCCGT

## Data Availability

The FASTQ files are available through NCBI Sequence Read Archive under project number PRJNA780617 and relative expression data are available through CrustyBase and in Appendix A.

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
