# Peer review of "Transcriptomic Analysis across Crayfish (Cherax quadricarinatus) Claw Regeneration Reveals Potential Stem Cell Sources for Cultivated Crustacean Meat"

_ijms, 2024, doi:10.3390/ijms25168623_

Round 1
Reviewer 1 Report
Comments and Suggestions for Authors
Major concerns,
1. I have a major concern. In this study, regeneration was divided into six stages, with stage 4 and 6 considered as potential sources of MSCs. However, these two stages are also in different molting stages. The gene expression differences observed in stages 4 and 6 could be due to molting rather than regeneration.
2. Sample s4.2 has a much higher R-value comparing to the other three samples in Stage 4. In contrast, samples in the other stages have relatively similar R-values. Does s4.2 affect the results of subsequent transcriptome analysis? If this sample is removed, will the results remain the same?
3. Wang et al. (2022) (DOI: 10.1126/sciadv.abl4642) studied the regeneration of Eriocheir sinensis and concluded that Innexin genes facilitate the early stage of regeneration. I am wondering how this gene is expressed in the current study. Could you compare the results of this study with those of Wang's study?
Author Response
Thank you very much for taking the time to review our manuscript. Your suggestions have been most helpful.
- I have a major concern. This study divided regeneration into six stages, with stages 4 and 6 considered potential sources of MSCs. However, these two stages are also in different molting stages. The gene expression differences observed in stages 4 and 6 could be due to molting rather than regeneration.
Thank you for raising this. Yes, molt stage is indeed a confounding factor in gene expression, however, this unfortunately can’t be avoided if comparing Stage 4 and Stage 6 in the current dataset. Your suggestion has however, prompted us to consider how this could be overcome in future. Potentially we could compare non-regenerating limb tissue just before and after the molt, to that of regenerating limb tissue at the same times. We’ve opted not to add this idea to the revised manuscript for the sake of brevity, however we do mention the broader issue of molt stage in the limitations section.
- Sample s4.2 has a much higher R-value comparing to the other three samples in Stage 4. In contrast, samples in the other stages have relatively similar R-values. Does s4.2 affect the results of subsequent transcriptome analysis? If this sample is removed, will the results remain the same?
Thank you for highlighting this. Yes, unfortunately, this is a very large contrast that has potentially skewed differential expression (DE) results. In fact, the one outlier in the PCA (Figure 2) is indeed sample S4.2. We have now revisited our DE data to compare expression values with and without the S4.2 sample. We found that its removal reduced the expression of most AM cuticle and blue pigmentation genes upregulated in Stage 4, alongside some metabolism related and muscle specific genes. Thus, it appears this outlier is largely responsible for the overexpression of AM cuticle synthesis activity in Stage 4 that was mentioned in the Discussion. We noted in our original submission that while cuticle synthesis is not directly related to growing muscle tissue in vitro its high expression here warranted mention. The presence and strong influence of the outlier has now been noted in Section 2.4 and the Discussion.
The main focus of the study was to compare activity of growth and cell cycle related genes. In revisiting the normalized gene counts for these genes after the removal of S4.2, we note no obvious differences in their expression.
- Wang et al. (2022) (DOI: 10.1126/sciadv.abl4642) studied the regeneration of Eriocheir sinensis and concluded that Innexin genes facilitate the early stage of regeneration. I am wondering how this gene is expressed in the current study. Could you compare the results of this study with those of Wang's study?
Thank you for alerting us to this important study. We have read it with interest and note a number of features potentially important to our future work. There are some major differences between our approach and theirs. Their model, the crab Eriocheir sinenis, displayed uniform regeneration after autotomy and they were able to sample tissues at 0, 1, 13 and 30 days post autotomy, the latter time point being the molt event. Our crayfish did not display the same uniformity and many took several months to molt. Thus, our sampling could not be as precise. We note however, that their second sample point (1dpa) is likely the most representative of our Stages 1 and 2, their third point (13dpa) representative of our Stages 3 and 4, and their fourth point similar to our Stage 5. With this in mind, we looked at the expression patterns they found, with particular attention to Innexin as you suggest. We found an Innexin2 in our dataset with the best homology match to Drosophila melanogaster Innexin, including an Innexin domain and an E-value of 7.56e-151. In contrast to Wang et al’s findings, it was not significantly upregulated in the earliest stages, but was quite uniformly expressed throughout our six stages. It therefore did not feature in our DEG set (See sections 2.5 and 4.8). There was however, like for many of the target genes, a trend towards upregulation in Stages 2 and 4, which is interesting. We have now addressed the Wang et al findings at the end of Section 3.1.
Reviewer 2 Report
Comments and Suggestions for Authors
Dear Authors,
This manuscript presents a significant advancement in cultivated meat research, particularly for crustaceans. Using transcriptomic analysis to identify potential stem cell sources from regenerating crayfish claws is innovative and has important implications for sustainable food production. The study is well-executed, and the findings are thoroughly analyzed and discussed. However, I have some suggestions for improvements:
Abstract:
Line 23 - 26 Include more specific quantitative results to give a clearer picture of the findings.
Line 18-20 this is a highlight key finding that needs to be written at the end of the Abstract
Introduction:
Line 44 Strengthen the justification by comparing Cherax quadricarinatus with other potential model organisms and explaining why it is particularly suitable for this research.
Line 79 – 82 Clearly state the specific research gap that this study aims to address instead of writing what and how your research was done.
Material and Methods:
Expand on Statistical Methods: Provide a more detailed description of the statistical methods used for differential expression analysis and GO term enrichment.
Discussion
Line 357-361 Start the discussion by clearly summarizing the most significant findings of the study to remind the reader of the key points before delving into their implications.
Line 545 At the end of the Discussion please elaborate on the practical implications of the findings for CCM production, including any challenges and potential solutions for translating these findings into a commercial setting.
Conclusion:
Line 751
Clearly and succinctly restate the most important results of the study, emphasizing the novel contributions to the field.
Elaborate on the practical applications and implications of the findings for the CCM industry, including how they can address current challenges.
Kind regards.
Comments on the Quality of English LanguageMinor editing of English language required.
Author Response
Thank you very much for taking the time to review our manuscript. Your suggestions have been most helpful.
Comments:
This manuscript presents a significant advancement in cultivated meat research, particularly for crustaceans. Using transcriptomic analysis to identify potential stem cell sources from regenerating crayfish claws is innovative and has important implications for sustainable food production. The study is well-executed, and the findings are thoroughly analyzed and discussed. However, I have some suggestions for improvements:
Abstract:
Line 23 - 26 Include more specific quantitative results to give a clearer picture of the findings.
Line 18-20 this is a highlight key finding that needs to be written at the end of the Abstract
Thank you. We have adjusted the abstract to incorporate these two helpful suggestions.
Introduction:
Line 44 Strengthen the justification by comparing Cherax quadricarinatus with other potential model organisms and explaining why it is particularly suitable for this research.
Thank you. We have included this in Section 1.4 (rather than at Line 44 which is in an unrelated section).
Line 79 – 82 Clearly state the specific research gap that this study aims to address instead of writing what and how your research was done.
Section 1.4 has also been updated to include this suggestion.
Material and Methods:
Expand on Statistical Methods: Provide a more detailed description of the statistical methods used for differential expression analysis and GO term enrichment.
Thank you. Sections 4.8 and 4.10 have been updated accordingly.
Discussion
Line 357-361 Start the discussion by clearly summarizing the most significant findings of the study to remind the reader of the key points before delving into their implications.
Thank you. The beginning of the Discussion has been updated.
Line 545 At the end of the Discussion please elaborate on the practical implications of the findings for CCM production, including any challenges and potential solutions for translating these findings into a commercial setting.
Thank you for this important suggestion. We have added a new Implications section to include this information (3.3) and included it in our updated Conclusion.
Conclusion:
Line 751
Clearly and succinctly restate the most important results of the study, emphasizing the novel contributions to the field.
Elaborate on the practical applications and implications of the findings for the CCM industry, including how they can address current challenges.
Thank you. The Conclusion has been amended accordingly.
Reviewer 3 Report
Comments and Suggestions for Authors
The authors present an impressive and mostly compelling study regarding the changes in gene expression during regenerating crayfish claws. Figure quality and experimental depth is suitable for publication. Inevitably, these types of transcriptomic comparative studies are vulnerable to overwhelming amounts of data and over-interpretation. At least the authors have narrowed their candidate gene lists down to a manageable numbers of transcripts and done some follow-up experiments, but the narrative seems unnecessarily long and over-speculative in places (e.g., broader categories of gene expression in Fig. 3, Table 5, etc.).
Also, the timeline of harvesting tissue is not obvious. It's clear from diagrams when the different stages were taken, but are these hours, days, months from each other? At what temperature were the crayfish grown?
The repetitive verbiage "expected" should be reduced, unless expectations can be supported with literature instead of assumptions.
Many of the Results are presented in the present tense, they should be switched to past tense accordingly.
The Conclusion is disproportionately short and should be expanded to include key candidate genes that the authors believe are critical to their underlying hypothesis and longer term goal (i.e., production of edible crustacean tissue in vitro), based on their collective data.
Line 377: "phenomena" doesn't seem to be the correct word here, but if so then it should be "phenomenon"
Comments on the Quality of English LanguageEnglish is adequate, some minor edits suggested
Author Response
Thank you very much for taking the time to review our manuscript. Your suggestions have been most helpful.
Comments and Suggestions for Authors
The authors present an impressive and mostly compelling study regarding the changes in gene expression during regenerating crayfish claws. Figure quality and experimental depth is suitable for publication. Inevitably, these types of transcriptomic comparative studies are vulnerable to overwhelming amounts of data and over-interpretation. At least the authors have narrowed their candidate gene lists down to a manageable numbers of transcripts and done some follow-up experiments, but the narrative seems unneessarily long and over-speculative in places (e.g., broader categories of gene expression in Fig. 3, Table 5, etc.).
Also, the timeline of harvesting tissue is not obvious. It's clear from diagrams when the different stages were taken, but are these hours, days, months from each other? At what temperature were the crayfish grown?
Thank you for raising this. Unlike in other crustacean models the individual regeneration time varied considerably between our crayfish depending on molt stage, primarily, but also age and individual health, all of which were unknown. In the Uca pugilator studies cited in the manuscript (refer to Section 2.1), multiple limb autotomy (MLA) was utilized to force ‘emergency’ molting. This enabled regeneration to occur immediately and synchronously allowing researchers to measure specific time periods for each stage. This was not possible in our case as we did not employ MLA and crayfish do not regenerate in a similar way. In our study the time taken to reach Stage 2 ranged from 26 to 60 days and the time to molt ranged from 42 to 138 days. I have added this information to Section 4.2 in the Methods.
The crayfish were kept at indoor ambient temperatures (21-24°C) during the months of November to May in Queensland. Water temperature was not measured. This has been added also.
The repetitive verbiage "expected" should be reduced, unless expectations can be supported with literature instead of assumptions.
Thank you. This has been reduced.
Many of the Results are presented in the present tense, they should be switched to past tense accordingly.
Corrected, thank you.
The Conclusion is disproportionately short and should be expanded to include key candidate genes that the authors believe are critical to their underlying hypothesis and longer term goal (i.e., production of edible crustacean tissue in vitro), based on their collective data.
This has been updated, thank you.
Line 377: "phenomena" doesn't seem to be the correct word here, but if so then it should be "phenomenon"
Corrected, thank you.